# Multi-Criteria Decision Methods for Selecting a Wind Farm Site Using a Geographic Information System (GIS)

Rahim Moltames [1], Mohammad Sajad Naghavi [1], Mahyar Silakhori [2,*], Younes Noorollahi [1], Hossein Yousefi [1,*], Mostafa Hajiaghaei-Keshteli [3] and Behzad Azizimehr [4]

[1] Energy Modelling and Sustainable Energy System (METSAP) Research Lab., Faculty of New Sciences and Technologies, University of Tehran, Tehran 1439956191, Iran
[2] Centre for Energy Technology, School of Mechanical Engineering, The University of Adelaide, Adelaide, SA 5005, Australia
[3] Tecnologico de Monterrey, Escuela de Ingeniería y Ciencias, Puebla 4500, Mexico
[4] Faculty of Engineering, Islamic Azad University West Tehran Branch, Tehran 1468763785, Iran
[*] Correspondence: mahyar.silakhori@adelaide.edu.au (M.S.); hosseinyousefi@ut.ac.ir (H.Y.)

**Abstract:** Wind energy is an economically, technically, and environmentally attractive option due to its cheapness and availability in different regions. The most important obstacle to developing renewable resources in Iran is subsidies for fossil fuels. The Iranian government has recently decided to reduce subsidies for electricity and fossil fuels, which has led to an increase in the prices of fossil fuels and electricity and makes renewable technologies more attractive to use. This study uses a multi-criteria decision method to identify wind energy potential in Khuzestan province. A GIS is used to determine the wind energy potential in this province. The technical, environmental, and economic criteria, which are a total of 14 layers of information, were examined by considering different values for each and from a Boolean point of view. The results show that, from the economic point of view, Shadegan city has the most potential and, from the technical point of view, Khorramshahr city has the highest amount of electricity production through wind energy. Furthermore, Dasht-e Azadegan city, due to its population, can supply the maximum amount of electricity it needs through wind energy. Among the three 550, 2500, and 8000 kW turbines, the 550 kW turbine has the most potential in the region.

**Keywords:** wind energy; GIS; site selection; multi-criteria decision method; windfarm

## 1. Introduction

Fossil fuels have disadvantages, such as the production of environmental pollutants, finitude, and high prices, which have led many governments to shift to alternative energy sources. One of the alternative energy sources is renewable energy sources that do not produce pollutants and are infinitely available in different parts of the world. Wind energy is one of these sources that has grown significantly in the past decades. This energy resource, like other renewable resources, has some limitations, such as oscillating production, dependence on topography, and environmental conditions, but with all these limitations, this energy source has several advantages that make it appealing to governments [1]. One of the great advantages of wind energy is the potential for the installation of wind turbines in locations far from electricity grids. It can also be used as a backup source of energy during peak hours [2–9]. According to the policies adopted by the Iran government, it is predicted that this energy will have a greater share in the future. Although wind energy has low economic costs and environmental problems and can reduce dependence on fossil fuel energy sources, it also has negative environmental impacts that should be studied using appropriate space policy tools to ensure that it aligns with national policies on infrastructural ecological and economic systems. For instance, during wind farm installation, new forest roads (or the widening of old roads) are opened to transport the necessary materials

and large machinery. The new roads have a significant negative influence on soil erosion and surface runoff [10,11]. A GIS has various industrial applications, and technological advancements have significantly enhanced GIS data, specifically, how they can be used and what can be achieved as a result. A GIS is a powerful decision-making tool for any business or industry, since it allows the analysis of environmental, demographic, and topographic data. Data intelligence compiled from GIS applications help companies and various industries, as well as consumers, make informed decisions. The GIS can identify the best area for a wind farm with the least amount of human error by identifying suitable potential areas using a combination of digital thematic maps and a conceptual model for data integration [12,13]. Identifying and selecting suitable sites for installing wind farms depends on various factors such as environmental, technical, geographical, and theoretical parameters. Choosing the right site for installing wind farms is one of the most important challenges in the development of wind resources [14]. Recently, a GIS has been widely used as a Decision Support System (DSS) to find a suitable location for the installation of wind farms [15–17]. The GIS as a DSS provides a large amount of spatial data in decision-making to evaluate and develop wind resources.

In this study, we use ArcGIS software to determine the potential of wind energy sources in the Khuzestan province in southwestern Iran. Important criteria such as technical, environmental, economic, and physiographic standards have been examined. This research paper divides the criteria into two parts, including restrictive and classified layers.

## 2. Literature Review

Several DSS methods have been used to identify the best wind farm installation site [18–20]. Rodman et al. identified a suitable wind farm site in northern California, USA, using a GIS model by considering physical, environmental, and socio-economic parameters [14]. In GIS tools, a set of influential technical, economic, and political parameters were used to determine the appropriate site for a wind farm in Iowa, USA [21]. A study was also conducted to determine the site of a hydro-plant installation in southwestern Taiwan [22]. Baban et al. considered 14 criteria, such as slope, location of historical sites, land use, etc., to find a suitable site for wind farm installation using weighted overlay analysis in a GIS [23]. Jank used various factors such as distance to roads and cities, wind potential, and population density using multi-criteria decision techniques in GIS tools to determine the appropriate location for a wind farm in Colorado [20]. Haran et al. added more parameters to Junk's study, such as economic costs for wind farm development [24]. Table 1 reviews the studies of other researchers and shows the layers of information each of them considered in finding the best site for wind farms.

**Table 1.** Summary of considered criteria from the literature review.

| Index | [25] | [26] | [27] | [21] | [28] | [29] | [14] | [30] | [23] | [31] | [32] |
|---|---|---|---|---|---|---|---|---|---|---|---|
| Slope | * | * | * | * | - | - | * | - | * | * | * |
| Distance from the airport | - | - | * | * | * | - | - | * | - | * | * |
| Distance from power transmission lines | - | - | * | * | * | - | - | * | * | - | * |
| Distance from the railway | * | * | - | * | * | - | - | - | - | - | - |
| Distance from protected areas | * | * | * | * | * | * | * | - | * | * | * |
| Distance from the lake | * | - | - | * | * | * | - | * | * | * | * |
| Distance from the main rivers | * | - | - | * | * | * | - | * | * | * | * |
| Distance from tributaries | * | - | - | * | * | * | - | * | * | * | * |
| Distance from the road | - | * | * | * | * | - | - | * | * | - | * |
| Distance from the city | * | * | * | * | * | * | * | * | * | * | * |
| Distance from the village | - | - | - | - | - | - | - | - | - | - | - |
| Distance from the dam | - | - | - | - | - | - | - | * | * | * | * |
| Land use | - | * | - | * | * | * | * | * | * | - | * |
| Wind speed | * | - | * | * | * | * | * | * | * | * | * |

(*) Considered, (-) Not considered.

Van Wick and Cooling [33] and Moriarty and Honri [34] proposed different wind energy potential measurement classifications and expressed the differences between the various potential groups. According to this research, there are five categories of potential groups for wind energy based on constraint factors: theoretical potential, geographical potential, technical potential, economic potential, and pure energy potential. Theoretical potential expresses the total wind energy regardless of existing constraints, such as economic, technical, etc. Geographical potential represents the amount of wind energy used by considering geographical constraints such as land usage. Technical potential indicates the amount of wind energy that can be extracted technically and from the specifications of the turbines. In the economic potential of the economic capacity of the project, future energy costs and future energy prices in the market are considered, and the amount of wind energy that is economically exploitable is shown. Pure energy potential expresses the total energy input to the turbine minus the energy wasted, which is the maximum energy extracted from the environment in a year. In this study, different groups of wind energy potential measurements are shown in Figure 1. Environmental potential includes the amount of wind energy extracted in terms of environmental constraints, such as distance from the forest, land usage, rivers, etc.

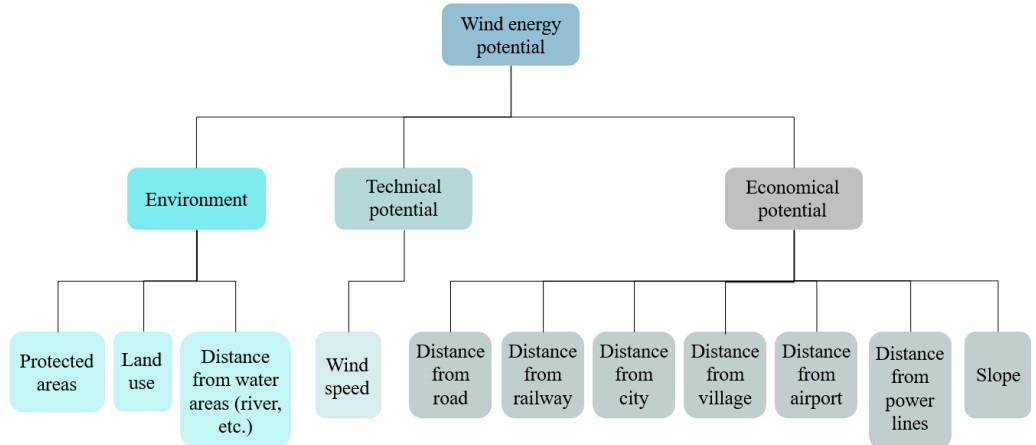

**Figure 1.** Different groups of wind energy potential measurements.

## 3. Materials and Methods

### 3.1. Case Study

The use of wind energy in Iran first started with the installation of two 500 kW wind farms in the Manjil and Rudbar regions in 1994; the average wind speed in the area near the Rudbar River was 15 m per s for 3700 h per year and also 13 m per s for 3400 h per year in Manjil region [35,36]. Currently, the capacity of wind farms in Manjil and Rudbar is 92.5 MW. In addition, a wind farm has been constructed in the Binalood mountain region in Khorasan province, which is in northeastern Iran, and the wind speed in this region is 9.8 m per s [35,36]. This hydro-plant has 45 wind turbines with a total capacity of 28.6 MW. We can also mention the wind farm constructed in the Zabol region in southeastern Iran, Shiraz in central Iran, Tabriz in the northwest, Takestan in the west, and Mahshahr in southern Iran. The capacity of all these wind turbines is in the range of 660–747 kW [36–38]. According to the wind atlas of Iran and considering the technical and economic potential of wind energy, it is estimated that the potential to generate electricity using wind energy in the country is about 15,000 MW [37–39]. Figure 2 shows a map of Iranian wind speed at 80 m.

Khuzestan province, with an area of 236.64 square km, between 47 degrees and 41 min to 50 degrees and 39 min east longitude of the Greenwich meridian and 29 degrees and 58 min to 33 degrees and 4 min north latitude of the equator, is located in southwestern Iran. This province borders Ilam province to the northwest, Lorestan province to the north, Chaharmahal-e-Bakhtiari, Kohgiluyeh, and Boyer-Ahmar provinces to the northeast and

east, the Persian Gulf to the south, and Iraq to the west. According to the country's division in 1375, this province has 15 cities, 35 districts, 13 villages, and 4496 settlements. Ahvaz is the capital of Khuzestan province. The population of this province is about 4.7 million people (2016), which is the fifth most populous province of Iran. Khuzestan province is surrounded by the Zagros Mountains to the north and east. Towards the province's center, the sea level decreases, and flat plains replace the high mountains. Khuzestan includes two mountainous and plain regions. Two-fifths of the total area of this province is mountainous, and three-fifths is plains. Kuh-e Chu, Zardkooh, Shavish, Abbandan, Mamazad, Kuh-e Siah, and Kuh-e Chal are among the mountains of Khuzestan. Khuzestan plain has a slight slope, and in some parts, there are salt domes that play a major role in the salinization of the land and water.

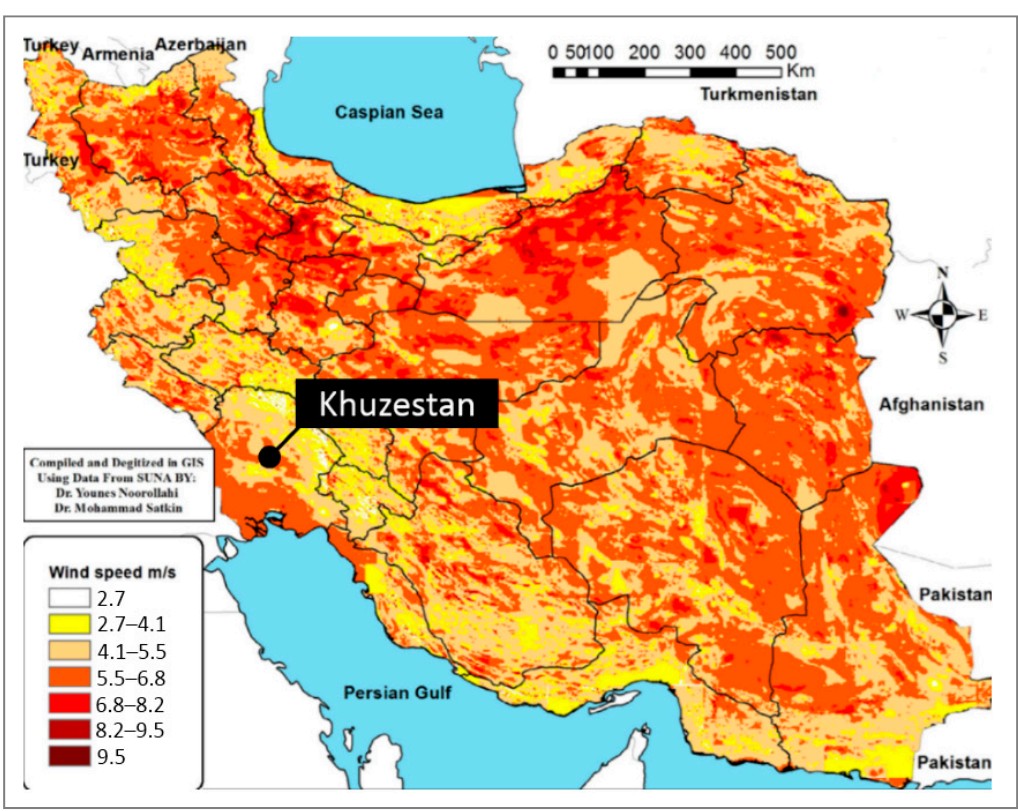

**Figure 2.** Map of wind speed in Iran at 80 m altitude [40].

Khuzestan is the most water-rich province of Iran. The five great rivers that originate from the Zagros irrigate the plains of Khuzestan and flow directly or indirectly into the Persian Gulf. Karun, the largest of these rivers, is the largest river in Iran, which originates from the Central Zagros mountains and joins the waters of Dez near Shushtar. Khuzestan province has different climates: a semi-desert climate that includes the cities of Abadan, Khorramshahr, Mahshahr, Hindijan, Azadegan plain, Dezful, Behbahan, Ramhormoz, Shushtar, and the northern regions of Ahvaz. The hot steppe climate includes the northern regions of Dezful, Behbahan, Ramhormoz, Shushtar, and northern Ahvaz. Khuzestan province is affected by three types of winds: the first wind is the cold flow of mountainous areas, and the second wind (sultry) is the hot and humid flow of the Persian Gulf that blows towards the plains. The third wind blows from Saudi Arabia and always brings some sand and moisture. Based on the data of synoptic stations in Khuzestan province in 1375, an absolute minimum temperature of $-0.2\ ^\circ$C and an absolute maximum temperature of $40.50\ ^\circ$C were reported in Ahvaz.

### 3.2. Selecting the Site of the Wind Farm

A GIS is "a computer-based system that can hold and use data describing surface locations of the earth" [41,42]. It is a tool that is widely used for decision making in various fields such as urban management, transportation planning, environmental management, telecommunications, service planning, national defense, network management, and marketing [43,44]. The methodology proposed is simple and provides rapid and accurate results at a low cost [45]. Depending on which person or organization uses this program, the definition and scope of its usage changes [46]. In a nutshell, this program is a combination of hardware, software, and procedure to facilitate management, analysis, and modeling to solve complex planning problems [47]. The main purpose of a GIS is to preserve geographical space data [44]. This program allows organized geographic data to be integrated with other data, and, as a result, a new series of data and useful information for decision making are created. One of the basic features of a GIS is data layers, which represent different characteristics in a specific area, allow the creation of specific criterion for each layer, and overlap all layers to produce an optimal area that meets all criteria. Therefore, it is widely used in site selection problems for various purposes [43].

Selecting a site for a large wind turbine requires the consideration of a comprehensive set of factors and a balance of several objectives. The selection of suitable project areas involves factors such as physical, demographic, economic, policy, and environmental factors. The selection criteria must also meet the optimistic criteria.

The Multi-Criteria Decision Making (MCDM) technique is adopted in different approaches to decision making. This method includes explicit statements of the decision maker's preferences. These preferences are represented by different values, scales, constraints, purposes, tools, and other parameters. They analyze and support the decision through a formal analysis of alternatives, their characteristics, evaluation criteria, objectives, and constraints. The multi-criteria decision method is used to solve various site selection problems.

The most important factor in the multi-criteria decision method is determining the "weight" of a set of criteria according to their importance. The Analytic Hierarchy Process (AHP) is a comprehensive, logical, and structural framework that allows the analyst to understand complex decisions by analyzing the problem in a hierarchical structure. Combining all the relevant decision criteria and comparing them in pairs allows the decision maker to determine the transactions between the objectives. The AHP allows decision makers to represent a complex problem in a hierarchical structure that illustrates the relationship of purpose, goals, criteria, and options.

### 3.3. Evaluation Process

The evaluation methodology is mainly based on three sources: country regulations, opinions of experts in this field, and findings from the literature reviews. Figure 3 shows the algorithm for this evaluation.

Wind speed: this is the most important factor in the selection of a site for a wind farm. There is a limitation for maximum and minimum wind speed, which is specified by the turbine manufacturer. The usual minimum and maximum wind speeds for current technologies with turbines are 6 and 25 m per s [48].

Slope: The land's slope is also one of the main factors for the proper operation of wind turbines and their installation. Slope affects the amount of power extracted from the turbine [48]. By increasing the slope of the land, the cost of building a wind farm increases. Therefore, the slope should be reasonable, which in this study is considered a maximum slope of 30%. The slope is calculated based on the province's digital elevation model.

Distance from city and village: A suitable distance from population centers is important: (1) Noise pollution of wind turbines; (2) Reduced wind speed: Small-scale artificial environments lead to climate change, which may cause up to 25% reduction in average wind speed in the installation areas; (3) Development of future residential sectors;

(4) Financial costs [49]; (5) Providing the required human resources at the location of the turbine.

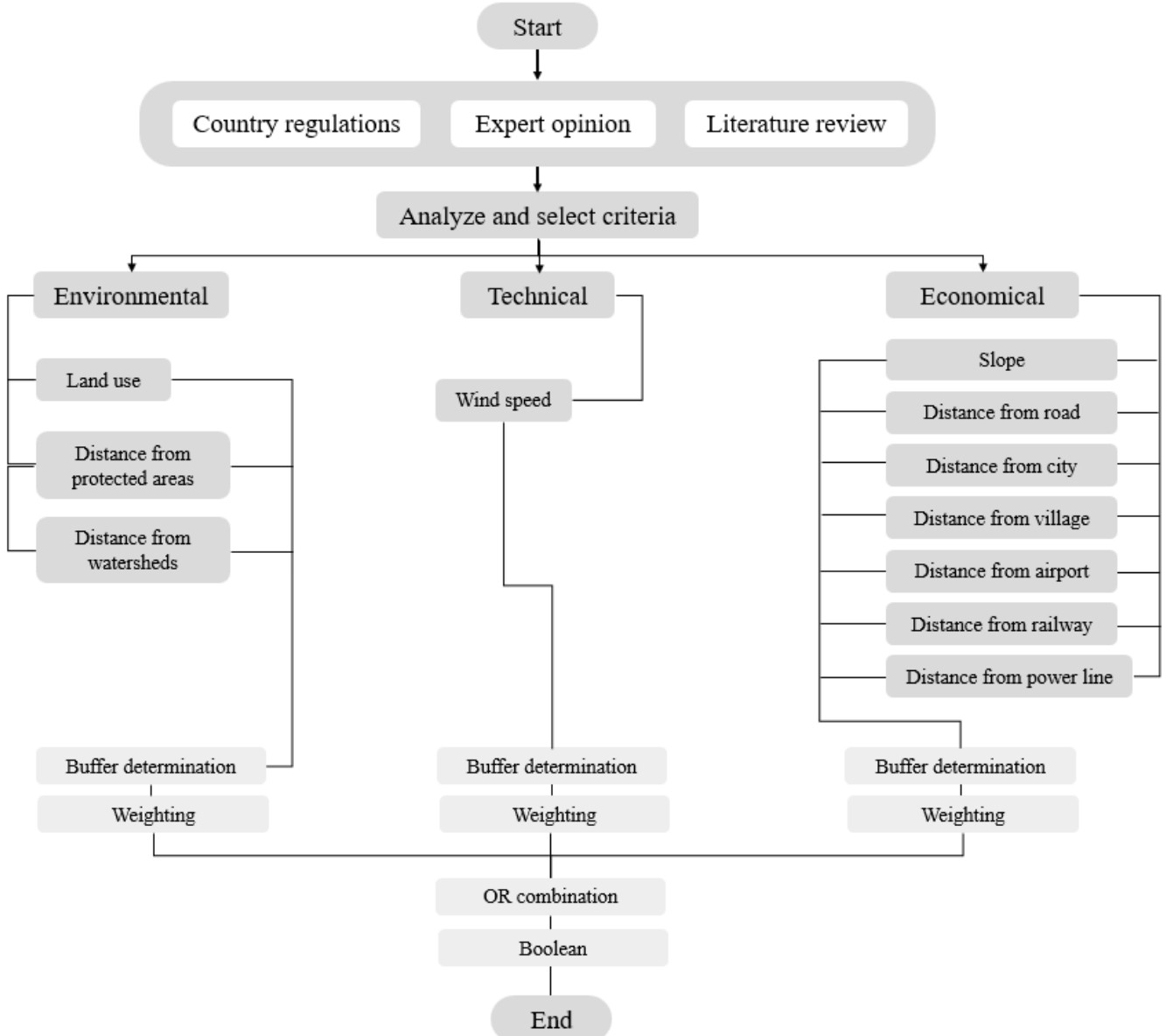

**Figure 3.** Hierarchical analysis of site selection of wind power plants.

Distance from rivers, lakes, and dams: Since the route of rivers is constantly changing and floods are likely to occur, the distance of wind farms from the riverbed increases the security of facilities [49].

Distance from routes: According to the regulations of each country, there is a standard distance from roads, which in Iran is 150 m. however, being close to transportation routes is economically important.

Distance from the airport: Wind turbines can interfere with the operation of air traffic control radars. In a radar system, aircraft are detected and tracked by changing the frequency of the return signal (Doppler Effect). Although the Doppler effect does not follow the position of wind turbines due to the absence of change, the orbit and rotation of the turbine wings cause the Doppler effect, which leads to the creation of interferences in tracking and disruption in aircraft identification [50].

Distance from environmentally protected areas: Wind farms hurt the inherent nature of these areas due to changes in natural landscapes and the creation of noise pollution. For this reason, wind turbines should be located at a suitable distance from protected areas.

Land use: It is not allowed to create a wind farm in specific environments, such as forests, lakes, and urban areas, to prevent any environmental damage.

To find suitable sites for wind farm installation, there are fourteen layers of input spatial data to overlap in ArcGIS 10.8 with GIS extension modules. Details of the input data are shown in Table 2.

**Table 2.** GIS layers.

| Index | GIS Data |
|---|---|
| Slope | Layer 1 |
| Distance from the airport | Layer 2 |
| Distance from power transmission lines | Layer 3 |
| Distance from the railway | Layer 4 |
| Distance from protected areas | Layer 5 |
| Distance from the lake | Layer 6 |
| Distance from the main rivers | Layer 7 |
| Distance from tributaries | Layer 8 |
| Distance from the road | Layer 9 |
| Distance from the city | Layer 10 |
| Distance from the village | Layer 11 |
| Distance from the dam | Layer 12 |
| Land use | Layer 13 |
| Wind speed | Layer 14 |

Constraints can be due to topographic factors that affect land usage planning; important factors related to topography include altitude and steep slopes. According to experts' ideas, installing wind farms in areas with high slopes, such as mountains and cliffs, increases investment. In addition to topographic factors, the country's rules and land use also create restrictions, for example, installation in residential and urban areas, and installation near rivers, airports, etc., all of which are required by law to have a standard distance. Once the constraints are determined based on Boolean logic, the CON tool is used to assign a true or false value (zero and one), and all areas covered by the restrictions are removed in the last step of the data analysis. These areas and their limitations are shown in Table 3.

**Table 3.** Elimination criteria and constraints.

| Reference | Constraint | Index | Layer |
|---|---|---|---|
| [51] | <30 | Slope (%) | Layer 1 |
| [52] | 5000 | Distance from the airport | Layer 2 |
| [52] | 250 | Distance from power transmission lines (m) | Layer 3 |
| [52] | 500 | Distance from the railway (m) | Layer 4 |
| Laws of Iran | 1500 | Distance from protected areas (m) | Layer 5 |
| Laws of Iran | 1500 | Distance from the lake (m) | Layer 6 |
| Laws of Iran | 750 | Distance from the main rivers (m) | Layer 7 |
| Laws of Iran | 750 | Distance from tributaries (m) | Layer 8 |
| [31] | 500 | Distance from the road (m) | Layer 9 |
| [31] | 2500 | Distance from the city (m) | Layer 10 |
| Laws of Iran | 1200 | Distance from the village (m) | Layer 11 |
| Laws of Iran | 1500 | Distance from the dam (m) | Layer 12 |
| [51] | 6 | Wind speed (m/s) | Layer 14 |

Each criterion is divided into relevant options using the AHP method, and its weight is calculated. First, the hierarchy structure of the criteria is formed according to the AHP method. Then, according to the degree of importance of the criteria to each other, the pairs

of factors were compared at each level, and, finally, the weights were calculated. Weighting scores for each criterion are obtained using an AHP. These weights are mostly obtained according to the conditions of the country and region under study. In this research, the highest weight is given to the wind speed, and the lowest is given to the distance from the main and secondary rivers. The weights of the layers are shown in Table 4.

**Table 4.** Weighing values for each criterion.

| Layer | Weight (%) |
|---|---|
| Layer 1 | 8 |
| Layer 2 | 8 |
| Layer 3 | 8 |
| Layer 4 | 6 |
| Layer 5 | 12 |
| Layer 6 | 4 |
| Layer 7 | 2 |
| Layer 8 | 2 |
| Layer 9 | 6 |
| Layer 10 | 11 |
| Layer 11 | 9 |
| Layer 12 | 4 |
| Layer 14 | 20 |

Valuation for each GIS data layer depends on its importance and appropriateness. The values and points of each layer, according to their unique characteristics and conditions prevailing in the province and the opinions of experts, are shown in Table 5.

**Table 5.** Valuation for each GIS data layer.

| Value | Slope ($^o$) | Distance from the Airport (km) | Distance from the Power Line (km) | Distance from the Railway (km) | Distance from Protected Areas (km) | Distance from the Lake (km) | Distance from the Main Rivers (km) | Distance from Tributaries (km) | Distance from the Road (km) | Distance from the City (km) | Distance from the Village (km) | Distance from the Dam (km) | Wind Speed (m/s) |
|---|---|---|---|---|---|---|---|---|---|---|---|---|---|---|
| 0 | >30 | 0–5 | 0–0.25 | 0–0.5 | 0–1.5 | 0–1.5 | 0–0.75 | 0–0.75 | 0–0.5 | 0–2.5 | 0–1.2 | 0–15 | 0–6 |
| 1 | 17–30 | 5–8 | 9–10 | 5–5.5 | 1.5–2 | 1.5–2 | 0.75–1.5 | 0.75–1.5 | 5–5.5 | 2.5–2.6 | 1.2–1.3 | 1.5–2 | - |
| 2 | 24–27 | 8–11 | 8–9 | 4.5–5 | 2–2.5 | 2–2.5 | 1.5–2.25 | 1.5–2.25 | 4.5–5 | 2.6–2.7 | 1.3–1.4 | 2–2.5 | 6–6.25 |
| 3 | 21–24 | 11–14 | 7–8 | 4–4.5 | 2.5–3 | 2.5–3 | 2.25–3 | 2.25–3 | 4–4.5 | 2.7–2.8 | 1.4–1.5 | 2.5–3 | - |
| 4 | 18–21 | 14–17 | 6–7 | 3.5–4 | 3–3.5 | 3–3.5 | 3–3.75 | 3–3.75 | 3.5–4 | 2.8–2.9 | 1.5–1.6 | 3–3.5 | 3.25–6.5 |
| 5 | 15–18 | 17–20 | 5–6 | 3–3.5 | 3.5–4 | 3.5–4 | 3.75–4.5 | 3.75–4.5 | 3–3.5 | 2.9–3 | 1.6–1.7 | 3.5–4 | - |
| 6 | 12–15 | 20–23 | 4–5 | 2.5–3 | 4–4.5 | 4–4.5 | 4.5–5.25 | 4.5–5.25 | 2.5–3 | 3–3.1 | 1.7–1.8 | 4–4.5 | 6.5–6.75 |
| 7 | 9–12 | 23–26 | 3–4 | 2–2.5 | 4.5–5 | 4.5–5 | 5.25–6 | 5.25–6 | 2–2.5 | 3.1–3.2 | 1.8–1.9 | 4.5–5 | - |
| 8 | 6–9 | 26–29 | 2–3 | 1.5–2 | 5–5.5 | 5–5.5 | 6–6.75 | 6–6.75 | 1.5–2 | 3.2–3.3 | 1.9–2 | 5–5.5 | 6.75–7 |
| 9 | 3–6 | 29–32 | 1–2 | 1–1.5 | 5.5–6 | 5.5–6 | 6.75–7.5 | 6.75–7.5 | 1–1.5 | 3.3–3.4 | 2–2.1 | 5.5–6 | - |
| 10 | 0–3 | 32–35 | 0.25–1 | 0.5–1 | 6–6.5 | 6–6.5 | 7.5–8.25 | 7.5–8.25 | 0.5–1 | 3.4–3.5 | 2.1–2.2 | 6–6.5 | >7 |

## 4. Results and Discussion

In this research, 14 layers of information, which include land slope, distance from the airport, railway line, road, urban and rural centers, main and secondary rivers, protected areas, and power transmission lines, were employed to select the best wind farm installation site in the Khuzestan province by using MCDM and a GIS. All mentioned layers were combined with specific weights and the decision was made based on Boolean logic.

### 4.1. AHP

Figure 4 shows the areas with limitations and the areas where it is possible to construct a wind farm. Figure 5 shows the weight percentage of these areas. The regions are classified into six categories according to their constraints and the weight ratios are given to each. Areas with a value of zero do not have the potential to construct wind farms in them, and areas with four to six are among the areas suitable for wind farms. After removing the

deprivation areas, it was determined that there could be the greatest potential for installing a wind farm in the southwestern part of the province (Figure 6).

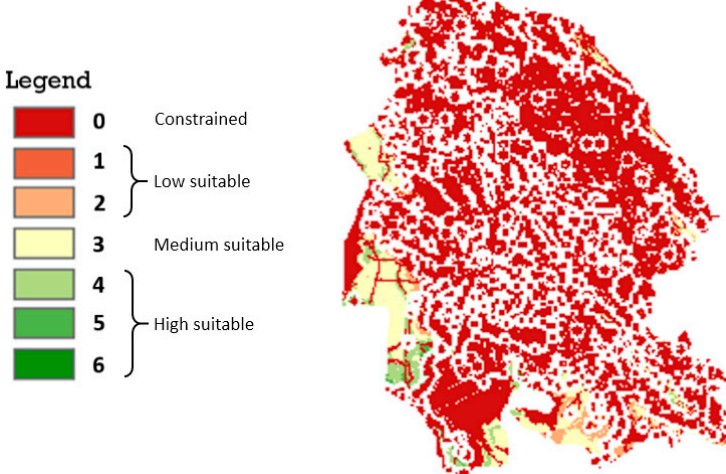

**Figure 4.** Different regions of the province by applying restrictions and values to each section.

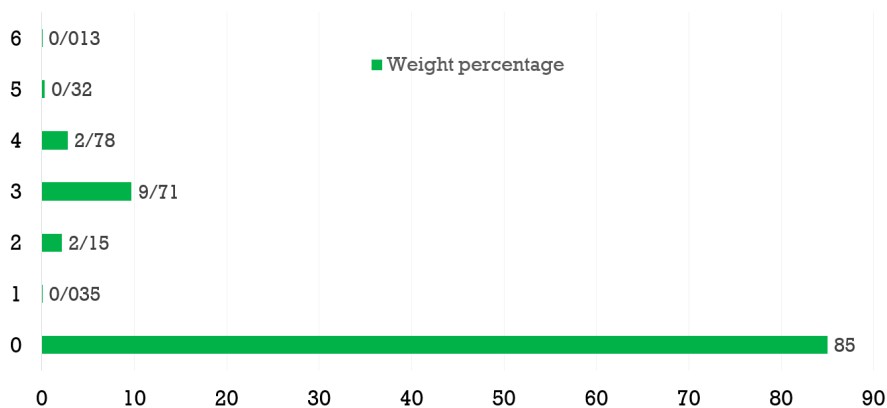

**Figure 5.** The weight percentages of different regions with constraints.

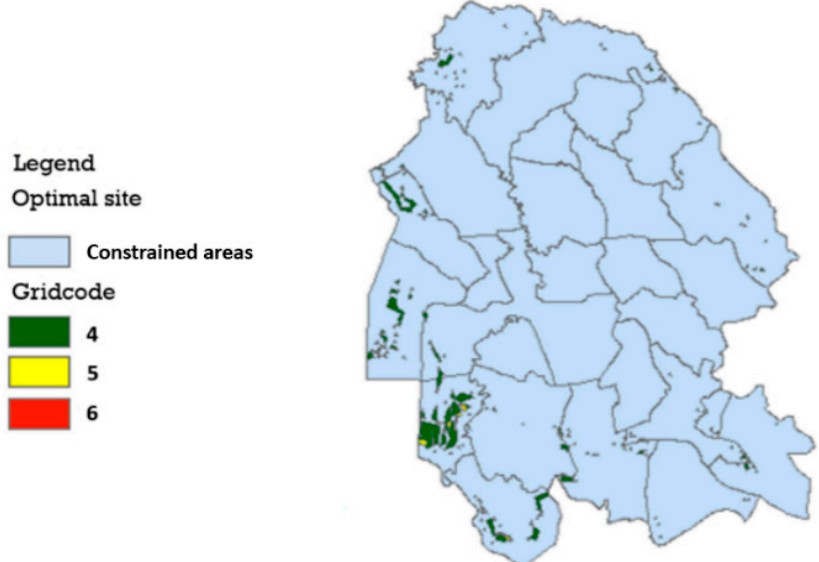

**Figure 6.** Areas with high potential for wind farm installation in Khuzestan.

It should be noted that only areas with high values (four to six) are considered appropriate areas, and other areas with medium, low, and limited values are not considered. Figure 6 shows the final output by applying all the constraints and applying the values and weights to each layer. Figure 7 also shows the weight percentage of suitable areas for wind farm installation.

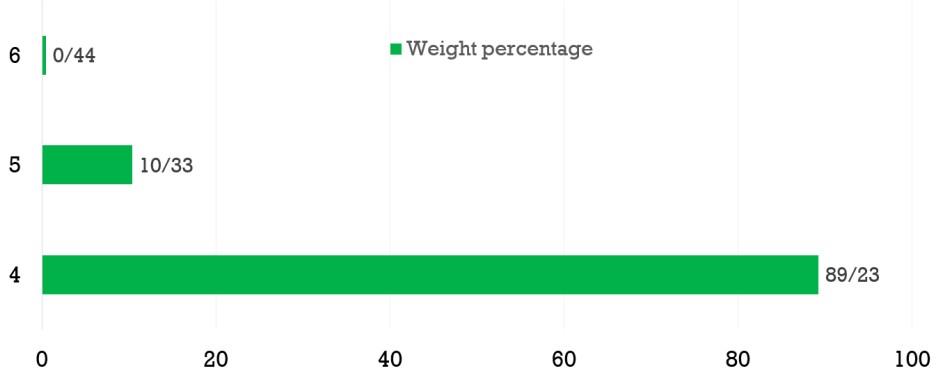

**Figure 7.** The weight percentage of suitable areas for wind farm installation.

The northeastern regions of the province have high altitudes and slopes and are considered mountainous areas, so these areas were considered restrictions. The central areas of the province were also considered limited due to the existence of urban and rural areas, rivers and watersheds, roads, and communication lines. The least restricted area is the southwestern part of the province. Wind speed is related to the type of ground cover and the ground slope. Forested areas and steep slopes have lower wind speeds than flat areas. According to the wind speed map (Figure 8) and the slope of the province (Figure 9), the northeastern regions of the province, due to their high slope, have lower wind speeds than the southwestern and central regions of the province; furthermore, in areas where rivers and forests are present, such as protected areas, the wind speed is low, which is due to the presence of forests, which reduces wind speed. In Figure 8, rivers are located on the wind speed map of the province with blue lines and protected areas with blue polygons, and, as can be seen, the margin of the southwestern regions of the province, which has the fewest rivers and forests, has the highest wind speed.

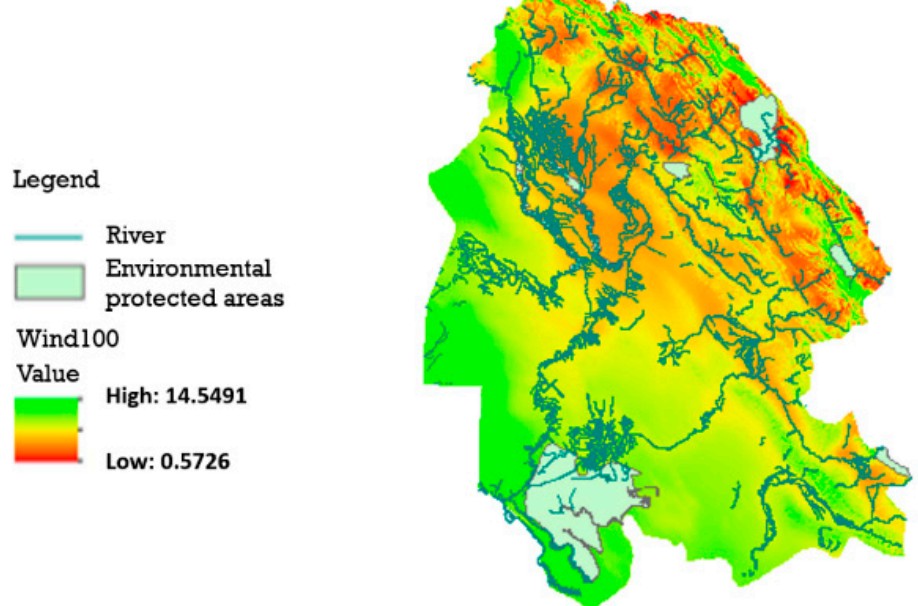

**Figure 8.** Wind speed map of the province together with protected areas and rivers.

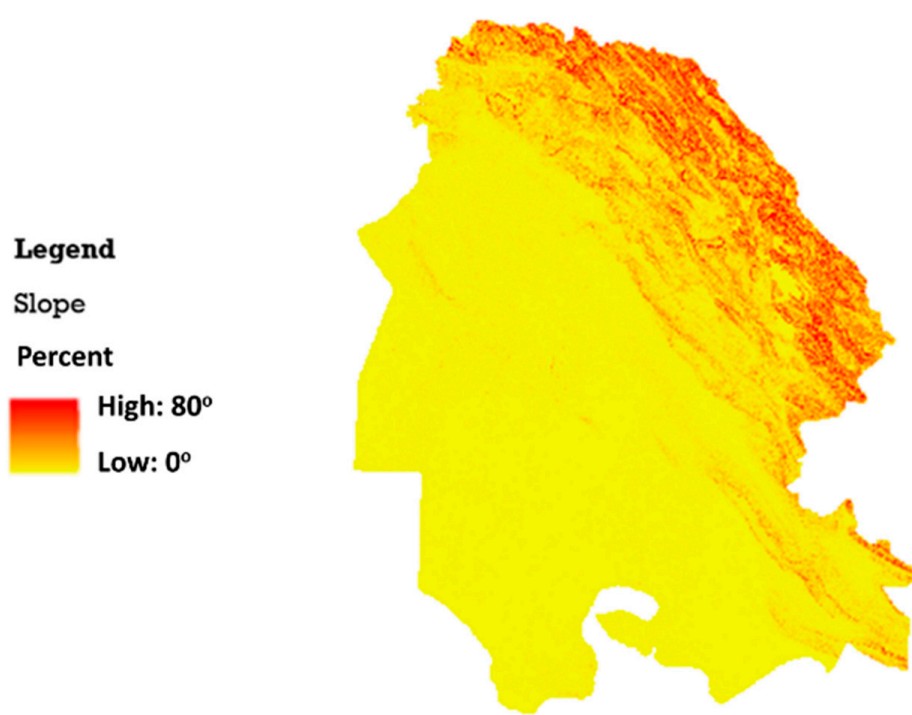

**Figure 9.** Slope map of Khuzestan province.

Locations with a logical distance from the urban areas are suitable for developing wind farms since they are closer to power transmission lines and transportation roads. These lands have a lower slope and are far enough from forest or rivers, and the predominant vegetation is grassland. Figure 10 shows the distribution of urban areas, as well as areas with the greatest potential for wind farms after the application of the constraints and weights of each layer. As can be seen, in the southwestern, western, and northwestern regions where urban centers are present, there is a high potential for the installation of wind farms, of course, considering the distance from them.

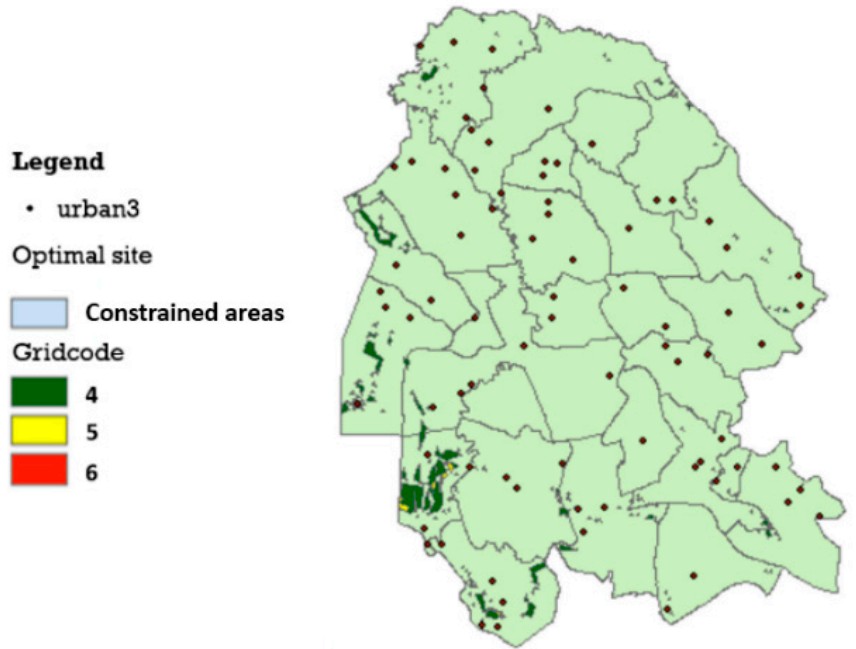

**Figure 10.** Areas with high potential for installation of wind farms in Khuzestan, considering urban areas.

The area of triple-potential areas for installing a wind farm is as follows (Figure 11). In total, about 116,198 hectares of land in Khuzestan province are suitable for installing a wind farm.

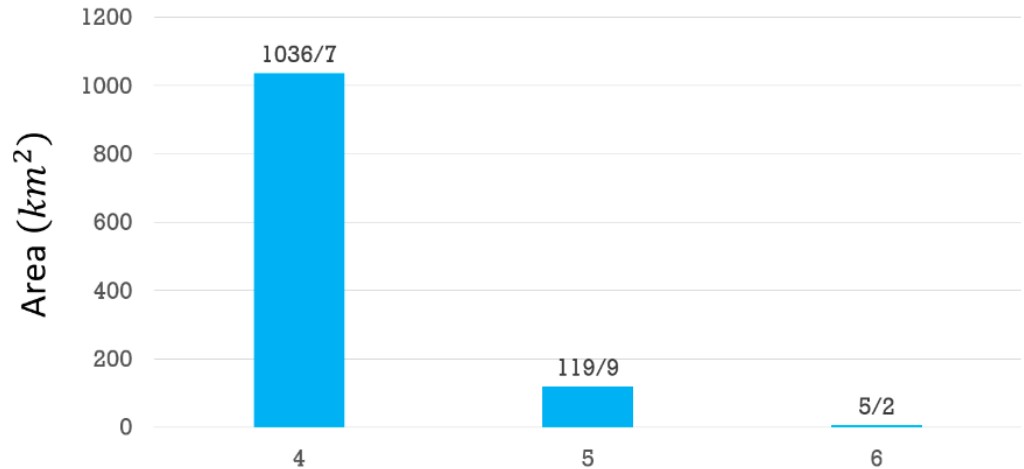

**Figure 11.** The area of triple-potential areas for the installation of a wind farm.

*4.2. Technical Potential*

In this study, three types of wind turbines with capacities of 550, 2500, and 8000 kW, (with rotor diameters of 41, 103.4, and 180 m, respectively) were considered. Several parameters were evaluated for each city of Khuzestan province, which were wind energy potential, the number of wind turbines, the amount of electricity generation through wind farms, and the percentage of electricity supply required in each city. The outcomes are summarized in Table 6.

**Table 6.** Information on suitable cities for the installation of wind farms in Khuzestan province.

| City | Suitable Areas (km²) | Population | Electricity Consumption (MWh) |
|---|---|---|---|
| Dasht-e Azadegan | 658.05 | 107,989 | 3,361,481 |
| Khorramshahr | 878.46 | 170,976 | 5,322,140 |
| Ahwaz | 100.39 | 1,302,591 | 40,547,052 |
| Abadan | 438.25 | 298,090 | 9,278,945 |
| Behbahan | 28.91 | 180,593 | 5,621,498 |
| Antiemetic | 63.17 | 171,412 | 5,335,712 |
| Dezful | 12.96 | 443,971 | 13,819,929 |
| Izeh | 27.63 | 198,871 | 6,190,456 |
| Shadegan | 23.87 | 138,480 | 4,310,605 |
| Bandar Mahshahr | 76.78 | 296,271 | 9,222,323 |
| Masjed Soleyman | 18.68 | 113,419 | 3,530,506 |
| Hendijan | 13.10 | 38,762 | 1,206,583 |
| Shush | 18.34 | 205,720 | 6,403,652 |
| Omidiyeh | 30.02 | 92,335 | 2,874,203 |

The results of using three types of wind turbines in potential cities are summarized in Tables 7–9. Figure 12 shows a comparison between the percentages of electricity required by potential cities supplied by wind turbines. Figure 13 also shows the amount of electricity generated through various turbines in potential cities.

According to Figures 12 and 13, Dasht-e Azadegan city has a higher demand for wind farms, and Khorramshahr city has the most potential for electricity generation through wind farms. Among the three types of wind turbines considered in this study, turbines with a capacity of 550 kW have the highest efficiency in potential locations. The main reason is the lower required area and, as a result, more turbines could be installed in a specific area.

It should be noted that the required land for each turbine, as well as the number of turbines used, is calculated by the following equation:

$$s = N \times a^2 \tag{1}$$

where *N* is the number of turbines and *a* is the rotor diameter of each turbine.

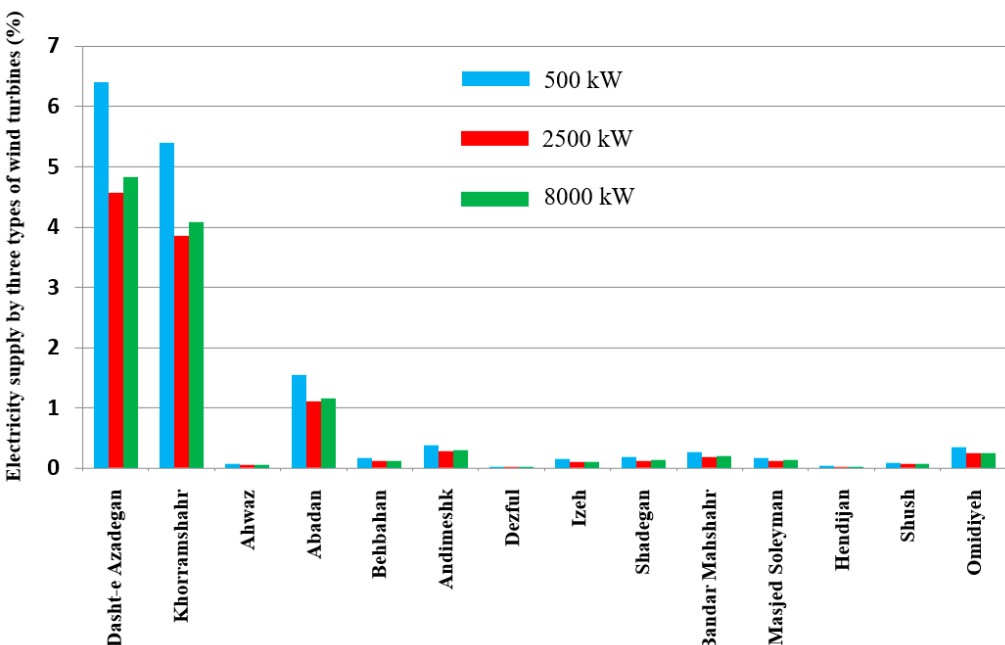

**Figure 12.** Percentage of electricity supply required by suitable cities through wind energy generated by three types of wind turbines.

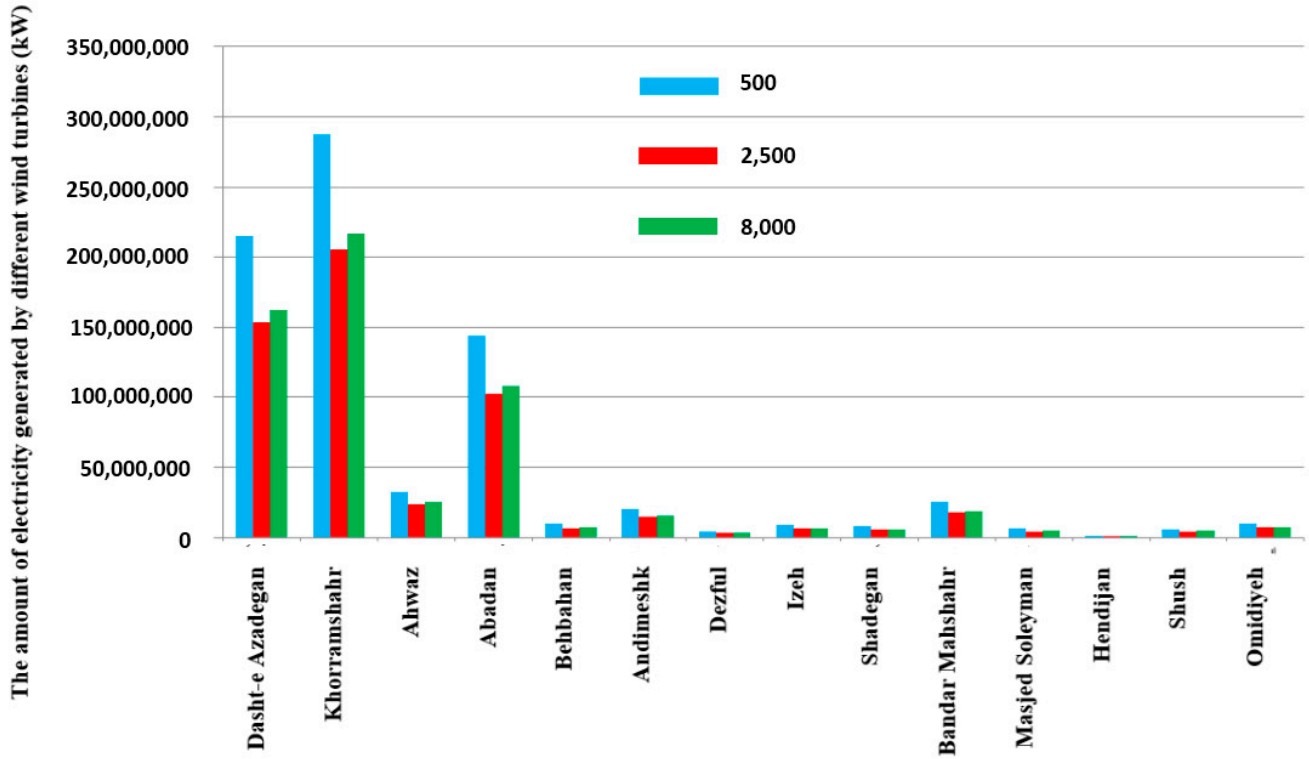

**Figure 13.** The amount of electricity generated by different wind turbines in suitable cities.

**Table 7.** Wind turbine with a capacity of 550 kW.

| City | Number of Turbines | Electricity Generated (MWh) | Percentage of Electricity Supplied through Wind Energy (%) |
|---|---|---|---|
| Dasht-e Azadegan | 391,467 | 215,306 | 6.41 |
| Khorramshahr | 522,583 | 287,420 | 5.40 |
| Ahwaz | 59,726 | 32,849 | 0.08 |
| Abadan | 260,708 | 143,389 | 1.54 |
| Behbahan | 17,203 | 9461 | 0.16 |
| Andimeshk | 37,580 | 20,668 | 0.38 |
| Dezful | 7710 | 4240 | 0.03 |
| Izeh | 16,440 | 9041 | 0.14 |
| Shadegan | 14,200 | 7810 | 0.18 |
| Bandar Mahshahr | 45,676 | 25,121 | 0.27 |
| Masjed Soleyman | 11,116 | 6113 | 0.17 |
| Hendijan | 780 | 428 | 0.03 |
| Shush | 10,915 | 6003 | 0.09 |
| Omidiyeh | 17,859 | 9822 | 0.34 |

**Table 8.** Wind turbine with a capacity of 2500 kW.

| City | Number of Turbines | Electricity Generated (MWh) | Percentage of Electricity Supplied through Wind Energy (%) |
|---|---|---|---|
| Dasht-e Azadegan | 61,549 | 153,872 | 4.57 |
| Khorramshahr | 82,164 | 205,410 | 3.85 |
| Ahwaz | 9391 | 23,476 | 0.05 |
| Abadan | 40,990 | 102,475 | 1.10 |
| Behbahan | 2705 | 6762 | 0.12 |
| Andimeshk | 5909 | 14,771 | 0.27 |
| Dezful | 1212 | 3030 | 0.02 |
| Izeh | 2585 | 6461 | 0.10 |
| Shadegan | 2233 | 5581 | 0.12 |
| Bandar Mahshahr | 7181 | 17,953 | 0.19 |
| Masjed Soleyman | 1748 | 4369 | 0.12 |
| Hendijan | 123 | 306 | 0.02 |
| Shush | 1716 | 4290 | 0.06 |
| Omidiyeh | 2808 | 7019 | 0.24 |

**Table 9.** Wind turbine with a capacity of 8000 kW.

| City | Number of Turbines | Electricity Generated (MWh) | Percentage of Electricity Supplied through Wind Energy (%) |
|---|---|---|---|
| Dasht-e Azadegan | 20,310 | 162,482 | 4.83 |
| Khorramshahr | 27,113 | 216,904 | 4.07 |
| Ahwaz | 3099 | 24,790 | 0.06 |
| Abadan | 13,526 | 108,210 | 1.16 |
| Behbahan | 893 | 7140 | 0.12 |
| Andimeshk | 1950 | 15,597 | 0.29 |
| Dezful | 400 | 3200 | 0.02 |
| Izeh | 853 | 6823 | 0.11 |
| Shadegan | 737 | 5893 | 0.13 |
| Bandar Mahshahr | 2370 | 18,958 | 0.20 |
| Masjed Soleyman | 577 | 4613 | 0.13 |
| Hendijan | 40 | 323 | 0.02 |
| Shush | 566 | 4530 | 0.07 |
| Omidiyeh | 927 | 7412 | 0.25 |

### 4.3. Economic Potential

In terms of economic potential, only areas with a value of six are considered. The four cities of Izeh, Shadegan, Abadan, and Khorramshahr, have the economic potential to build a wind farm. In regard to economic potential, three types of turbines are investigated. The characteristics of economically suitable areas are shown in Table 10.

**Table 10.** Economically viable cities for wind farm installation.

| City | Potential Area ($m^2$) | Electricity Consumption per City (MWh) |
|---|---|---|
| Izeh | 856,053.26 | 6,190,456 |
| Abadan | 861,079.23 | 9,278,945 |
| Shadegan | 892,226.11 | 4,310,605 |
| Khorramshahr | 872,843.19 | 5,322,140 |

If 550, 2500, and 8000 kW turbines are used, the results are shown in Tables 11–13.

**Table 11.** Results for 550 kW wind turbine.

| City | Number of Turbines | Generated Electricity (MWh) | Percentage of Electricity Supplied through Wind Energy (%) |
|---|---|---|---|
| Izeh | 509 | 280 | 0.004 |
| Abadan | 512 | 281 | 0.003 |
| Shadegan | 531 | 291 | 0.006 |
| Khorramshahr | 519 | 285 | 0.005 |

**Table 12.** Results for 2500 kW wind turbine.

| City | Number of Turbines | Generated Electricity (MWh) | Percentage of Electricity Supplied through Wind Energy (%) |
|---|---|---|---|
| Izeh | 80 | 200 | 0.003 |
| Abadan | 81 | 201 | 0.002 |
| Shadegan | 83 | 208 | 0.004 |
| Khorramshahr | 82 | 204 | 0.003 |

**Table 13.** Results for 8000 kW wind turbine.

| City | Number of Turbines | Generated Electricity (MWh) | Percentage of Electricity Supplied through Wind Energy (%) |
|---|---|---|---|
| Izeh | 26 | 211 | 0.003 |
| Abadan | 27 | 212 | 0.002 |
| Shadegan | 28 | 220 | 0.005 |
| Khorramshahr | 27 | 215 | 0.004 |

The amount of electricity generated by different turbines in potential cities and the percentage of electricity required by potential cities that are supplied by wind energy using different turbines are shown in Figures 14 and 15, respectively.

According to the above figures, Shadegan city has the greatest economic potential for installing a wind farm, and 550 kW wind turbines are the most efficient for this city.

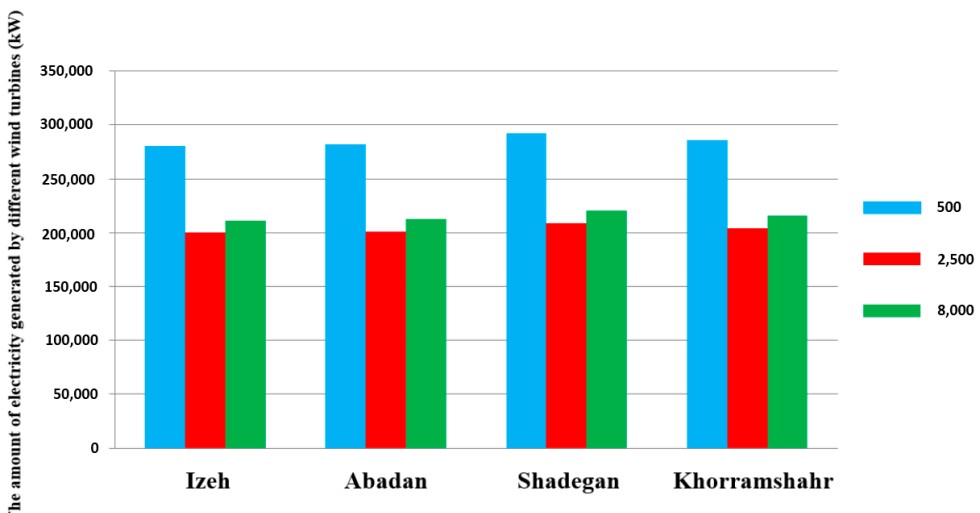

**Figure 14.** The amount of electricity generated by different power plants in potential cities.

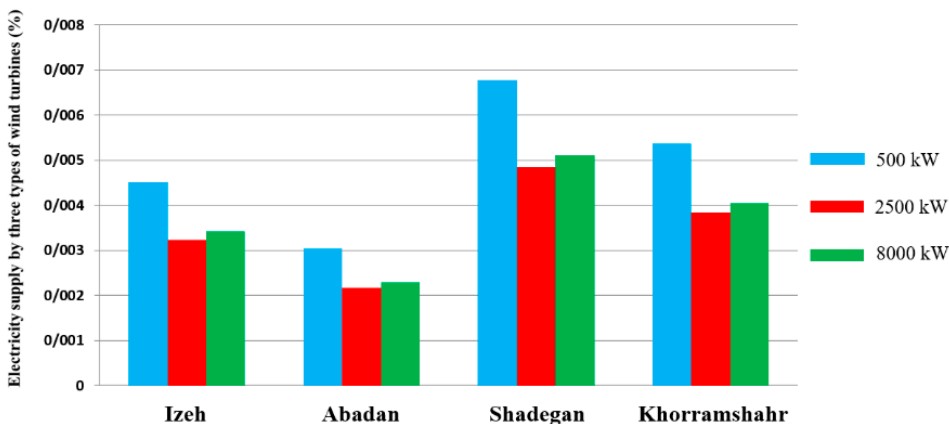

**Figure 15.** Percentage of electricity supply required by potential cities through wind energy generated by different wind turbines.

### 5. Conclusions

In this study, a multi-criteria decision method is used to determine the potential of wind energy sources in Khuzestan province. For this purpose, ArcGIS software was used. First, a multi-criteria decision method was studied to evaluate the region's wind resources. The technical, environmental, and economic criteria, which consist of 14 layers of information, were examined by considering different values for each of them and from a Boolean perspective. Furthermore, this research paper divided the criteria into two parts, including restrictive and classified layers. The results show that:

- From the economic point of view, Shadegan city has the most potential, as well as from the technical point of view.
- Khorramshahr city has the highest amount of electricity production through wind energy.
- Dasht-e Azadegan city, due to its population, can receive the maximum amount of electricity from wind energy.

There are some limitations in the applicability of the GIS data, which are mainly related to inconsistencies in the data, the lack of standardization, and the lack of comparable data across geographic areas. These limitations need to be considered in the results extracted from the analysis to maintain the accuracy and acceptability of the decisions.

**Author Contributions:** Conceptualization, Y.N., H.Y. and M.H.-K.; methodology and software, R.M. and M.S.N.; resources and data curation, B.A.; writing—original draft preparation, R.M. and B.A.; writing—review and editing, M.H.-K., M.S.N. and M.S.; supervision, Y.N. and H.Y. All authors have read and agreed to the published version of the manuscript.

**Funding:** This research received no external funding.

**Institutional Review Board Statement:** Not applicable.

**Informed Consent Statement:** Note applicable.

**Data Availability Statement:** Not applicable.

**Conflicts of Interest:** The authors declare no conflict of interest.

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
