# Peer review of "Multi-Criteria Decision Methods for Selecting a Wind Farm Site Using a Geographic Information System (GIS)"

_sustainability, doi:10.3390/su142214742_

Round 1
Reviewer 1 Report
Dear Editor.
I have finished my review on the proposed paper “Multi-criteria Decision Methods for Selecting a Wind-farm Site Using Geographic Information System (GIS) in Khuzestan”, sustainability-1896533-peer-review-v1.
Summary of the manuscript:
In the proposed paper, the authors’ goal is to identify priority areas for the development of wind energy potential in Khuzestan province of Iran. The authors used Multi-criteria Decision Methods, Boolean logic and GIS in order to find the most appropriate areas for the establishment of wind turbines. The technical, environmental, and 20 economic criteria, which are a total of 14 layers of information, have been examined by considering 21 different values for each and from Boolean's point of view.
General review:
1. Generally, the manuscript presents an interesting topic and the specific research seems to include some significant points for the research community of this field. The adding value of this paper is that the research subject is very significant in the frame of climate change.
2. The proposed paper is very well written with very good use of English language. Except some very minor grammatical mistakes and word errors. The authors should check again the paper to correct these minor mistakes.
3. The proposed paper is strangely structured. It begins with the Introduction with some references that helps the reader to get into the subject. In Introduction there is an effort to provide previous studies with similar scientific content, which took place in the research area and in other countries. At the end of Introduction, authors clearly state the goals of the research. However, the authors have used sub-sections in Introduction, which is very unusual. See, specific comments below.
4. The methodology is generally interesting, and explained, so other researchers could easily repeat it. However, some parts need to be clearer. See below specific comments.
5. The results are generally OK. However, some change should be made.
6. The quality of the work in Discussion is not adequate. See below specific comments.
7. Conclusions are appropriate for this paper.
Additional points for revision:
In my opinion, the proposed paper could be characterized as a good research work, complies with aims of Sustainability.
INTRODUCTION: This part of the study is very unusual for research study. You used 3 sub-sections in Introduction. However, I think that most of this information should be added to Methodology section. Also, I don not think that the MDPI allows sub-sections in Introduction.
Sections: 2. Case study, 3. Selecting the Site of the Wind Farm. I think that all these sections should be sub-sections under the methodology section. Generally, the research studies have 4 main sections: 1. Introduction, 2. Materials and Methods, 3. Results and Discussion, 4. Conclusions. These main sections are those that MDPI uses as template.
Line 54: DSS… this is the first time that DSS appears in the text. Please refer the complete acronym.
Figure 1: Please, show in the map where is the Khuzestan region.
Lines 181-182: Here, you say a few words about the GIS and land suitability assessment using multiple criteria. You should add some more lines and provide some more aspects of using MCDM in studies with various scopes. I proposed to add the following studies and take into account the reference lists of these studies. These studies used multiple criteria, Boolean logic and GIS for various purposes (Tzioutzios et al 2020 https://doi.org/10.3390/ijgi9120725, Van der Horst and Gimona https://doi.org/10.1016/j.biocon.2004.11.020, Eastman 1999).
266-268: Which are these studies. Please, add some literature.
Line 297: The word “susceptible” I think is wrong. May be is better “appropriate”?
Table 6: This table is meaningless, because shows the same information with figure 4. In figure 4 in the legend add the respective suitability classes form table 6.
Figure 6: Change the default “all other values” with “Constrained areas”. The same in figure 10.
Figure 9: In legend replace the word “value” and add the perspective measurement unit (% or else).
DISCUSSION: There is no discussion of the results. Discussion within the context of comparing the results of the paper with other studies, in not exists. I searched the paper, but I did not find references. You should compare your results with previously published studies.
References
Eastman, R. 1999. Multi-criteria evaluation in GIS. In Geographical Information Systems; Longley, P.A., Goodchild, M.F., Maguire, D.J., Rhind, D.W., Eds.; Whiley: New York, NY, USA, 1999; Chapter 35; pp. 493–502.
Van der Horst, D.; Gimona, A. Where new farm woodlands support biodiversity action plans: A spatial multi-criteria analysis. Biol. Conserv. 2005, 123, 421–432.
Tzioutzios, C.; et al. A. Multi-Criteria Evaluation (MCE) Method for the Management of Woodland Plantations in Floodplain Areas. ISPRS Int. J. Geo-Inf. 2020, 9, 725.
Author Response
Authors' Response to the Review Comments
Journal: Sustainability
Manuscript ID: sustainability-1896533
Paper title: Multi-criteria Decision Methods for Selecting a Wind-farm Site Using Geographic Information System (GIS) in Khuzestan
Authors: Rahim Moltames, Mohammad Sajad Naghavi, Mahyar Silakhori, Younes Noorollahi, Hossein Yousefi, Mo-stafa Hajiaghaei-keshteli, Behzad Azizimehr
Corresponding Authors: Hossein Yousefi (Email: hosseinyousefi@ut.ac.ir), Mahyar Silakhori (Email: mahyar.silakhori@adelaide.edu.au)
The authors would like to thank the area editor and the reviewers for their precious time and invaluable comments, which helped us to improve the quality of this manuscript. We have carefully considered and addressed all comments and the corresponding changes and refinements are summarized in our response below (the changes and corrections made in the revised manuscript are in GREEN):
Reviewer 1:
- INTRODUCTION: This part of the study is very unusual for research study. You used 3 sub-sections in Introduction. However, I think that most of this information should be added to Methodology section. Also, I don not think that the MDPI allows sub-sections in Introduction.
Reply: Thanks for your valuable comment. We changed the structure of this section.
- Sections: 2. Case study, 3. Selecting the Site of the Wind Farm. I think that all these sections should be sub-sections under the methodology section. Generally, the research studies have 4 main sections: 1. Introduction, 2. Materials and Methods, 3. Results and Discussion, 4. Conclusions. These main sections are those that MDPI uses as template.
Reply: Thanks for your valuable comment. We changed the structure of the sections.
- Line 54: DSS… this is the first time that DSS appears in the text. Please refer the complete acronym.
Reply: Thanks for your valuable comment. We added the complete description of the abbreviation.
- Figure 1: Please, show in the map where is the Khuzestan region.
Reply: Thanks for your valuable comment. We added a point showing the mentioned province inside the map.
- Lines 181-182: Here, you say a few words about the GIS and land suitability assessment using multiple criteria. You should add some more lines and provide some more aspects of using MCDM in studies with various scopes. I proposed to add the following studies and take into account the reference lists of these studies. These studies used multiple criteria, Boolean logic and GIS for various purposes (Tzioutzios et al 2020 https://doi.org/10.3390/ijgi9120725, Van der Horst and Gimona https://doi.org/10.1016/j.biocon.2004.11.020, Eastman 1999).
Reply: Thanks for your valuable comment. References added to the text.
- 266-268: Which are these studies. Please, add some literature.
Reply: Thanks for your valuable comment. Some papers are added to the revised manuscript.
- Line 297: The word “susceptible” I think is wrong. May be is better “appropriate”?
Reply: Thanks for your valuable comment. We changed the word according to your suggestion.
- Table 6: This table is meaningless, because shows the same information with figure 4. In figure 4 in the legend add the respective suitability classes form table 6.
Reply: Thank you. We removed the table and edited the figure in accordance with your suggestion.
- Figure 6: Change the default “all other values” with “Constrained areas”. The same in figure 10.
Reply: Thank you. We changed the words according to your suggestion.
- Figure 9: In legend replace the word “value” and add the perspective measurement unit (% or else).
Reply: Thank you. The Figure is updated.
- DISCUSSION: There is no discussion of the results. Discussion within the context of comparing the results of the paper with other studies, in not exists. I searched the paper, but I did not find references. You should compare your results with previously published studies.
Reply: Thanks for your true comment. We already searched out all the existing sources in scientific data bases and even governmental reports. We did not find any other study with similar concept to be able to have an apple-to-apple comparison between the results. It is actually one of the aims of this paper to provide a trustable analytical source to pave the way for further research or commercial actions.

Reviewer 2 Report
I have no additional comments.
Author Response
Authors' Response to the Review Comments
Journal: Sustainability
Manuscript ID: sustainability-1896533
Paper title: Multi-criteria Decision Methods for Selecting a Wind-farm Site Using Geographic Information System (GIS) in Khuzestan
Authors: Rahim Moltames, Mohammad Sajad Naghavi, Mahyar Silakhori, Younes Noorollahi, Hossein Yousefi, Mo-stafa Hajiaghaei-keshteli, Behzad Azizimehr
Corresponding Authors: Hossein Yousefi (Email: hosseinyousefi@ut.ac.ir), Mahyar Silakhori (Email: mahyar.silakhori@adelaide.edu.au)
The authors would like to thank the area editor and the reviewers for their precious time and invaluable comments, which helped us to improve the quality of this manuscript. We have carefully considered and addressed all comments and the corresponding changes and refinements are summarized in our response below (the changes and corrections made in the revised manuscript are in GREEN):
Reviewer 2:
I have no additional comments.
Reply: Thank you for your kind response.

Reviewer 3 Report
The manusript aims to create a methodology for wind Farm siting at a specific area using MCDM like AHP and GIS.
In this regard, the work has an original value. It has the capacity to contribute to the efforts made in this regard. However, it should review some basic considerations. After these improvements, it is acceptable for publication in your Journal.
- Please consider editing the title to exclude the geographic reference. Given the international scope of the journal, specific place names are discouraged from manuscript titles. Focus instead on methods and approaches in the title.
- I suggest that the authors present the literature review in a separate section.
- Section 3 "Selecting the Site of the Wind Farm" must be restructured, sub-sections to be redone
- What GIS software used in this work, the explanation of the choice I think very useful
- The presentation of the conclusion were not enough; it should be highlighted.
- The authors should elaborate more on the practical implications of their study, as well as the limitations of the study.
-As per the Turnitin software, the similarity index is 35% must be improved.
Author Response
The authors would like to thank the area editor and the reviewers for their precious time and invaluable comments, which helped us to improve the quality of this manuscript. We have carefully considered and addressed all comments and the corresponding changes and refinements are summarized in our response below (the changes and corrections made in the revised manuscript are in GREEN):
Reviewer 3:
- Please consider editing the title to exclude the geographic reference. Given the international scope of the journal, specific place names are discouraged from manuscript titles. Focus instead on methods and approaches in the title.
Reply: Thank you. We changed the title according to your suggestion.
- I suggest that the authors present the literature review in a separate section.
Reply: Thank you. We added the “Literature review” section to the revised manuscript.
- Section 3 "Selecting the Site of the Wind Farm" must be restructured, sub-sections to be redone.
Reply: Thank you. We changed the structure of the sections.
- What GIS software used in this work, the explanation of the choice I think very useful.
Reply: Thanks for your true comment. We missed the name of the software in the manuscript. It is ArcGIS software, which is one of the pioneer software in this field. The name is added to the manuscript.
- The presentation of the conclusion was not enough; it should be highlighted.
Reply: Thanks for your comment. The form of the presentation of the conclusion is revised.
- The authors should elaborate more on the practical implications of their study, as well as the limitations of the study.
Reply: Thanks for your comment. We improved the sentences related to this comment in the manuscript.
- As per the Turnitin software, the similarity index is 35% must be improved.
Reply: Thank you to highlight this important point. We revised the manuscript. The current similarity index is 14%. The report is attached.

Round 2
Reviewer 1 Report
Dear authors
Thank you very much for the provided responses. I believe that the paper has significant;y improved. I studied all the provided documents and you have done good job in review process. However, I have some minor comments to add. First, in lines 44-46 (new version) you say about "the negative environmental impacts of wind energy". I agree with you, and it is well known that during the wind farm installation, new forest roads (or widening of old roads) are opened in order to transport the necessary materials and large machinery. The new roads have significant negative influence on soil erosion and and surface runoff. For that reason, I recommend to add a small phrase here about the roads, and adding the two following studies to support the phrase (https://doi.org/10.3390/f11111201 and https://doi.org/10.15244/pjoes/81615). I believe that the writing style and english are fine, and just a final check is required.
Author Response
The authors would like to thank again the area editor and the reviewers for their precious time and invaluable comments. The changes and corrections made in the revised manuscript are in GREEN):
Reviewer 1:
Thank you very much for the provided responses. I believe that the paper has significant;y improved. I studied all the provided documents and you have done good job in review process. However, I have some minor comments to add. First, in lines 44-46 (new version) you say about "the negative environmental impacts of wind energy". I agree with you, and it is well known that during the wind farm installation, new forest roads (or widening of old roads) are opened in order to transport the necessary materials and large machinery. The new roads have significant negative influence on soil erosion and and surface runoff. For that reason, I recommend to add a small phrase here about the roads, and adding the two following studies to support the phrase (https://doi.org/10.3390/f11111201 and https://doi.org/10.15244/pjoes/81615). I believe that the writing style and English are fine, and just a final check is required.
Reply: Thank you for your suggestion. The corrections are made in the revised manuscript.

Reviewer 3 Report
Thank you for considering almost suggestions but the authors also should elaborate more on the practical implications of their study, as well as the limitations of the study, and further research opportunitiesin conclusion.
Author Response
Reviewer 3:
Thank you for considering almost suggestions but the authors also should elaborate more on the practical implications of their study, as well as the limitations of the study, and further research opportunities in conclusion.
Reply: Thank you for your suggestion. The corrections are made in the revised manuscript.
